# Effects of interferential stimulation on clinical symptom and urodynamic findings in women with voiding dysfunction: A protocol of randomized clinical trial

Seyedeh Saeideh Babazadeh-Zavieh[1,2], Fahimeh Karshenas[1], Sona Tayebi[3], Niloofar Rabiei[4], Seyed Mohammad Jafar Haeri ![ORCID][5]*

**1** Iranian Center of Excellence in Physiotherapy, Rehabilitation Research Center, Department of Physiotherapy, School of Rehabilitation Sciences, Iran University of Medical Sciences, Tehran, Iran, **2** Department of Physical Therapy, School of Rehabilitation, Arak University of Medical Sciences, Arak, Iran, **3** Department of Urology, Faculty of Medicine, Iran University of Medical Sciences, Tehran, Iran, **4** Department of Biostatistics, School of Public Health, Hamadan University of Medical Sciences, Hamadan, Iran, **5** Department of Anatomical Sciences, School of Medicine, Arak University of Medical Sciences, Arak, Iran

\* haeri1982@gmail.com

## Abstract

### Background

Voiding dysfunction, characterized by abnormally slow and/or incomplete micturition, is a clinical challenge that significantly impacts quality of life among women. Dysfunctional voiding, a subtype of voiding dysfunction is identified by intermittent and/or fluctuating urine flow caused by involuntary periurethral striated muscle contractions during voiding in individuals without neurological abnormalities. Interferential stimulation is a non-invasive surface electrical stimulation modality widely employed in physiotherapy. The current evidence supports its clinical efficacy in managing both urinary and fecal disorders. However, its role in treating dysfunctional voiding in women remains understudied. Thus, we present a study protocol to investigate the effects of interferential electrical stimulation on clinical symptoms and urodynamic findings in women with dysfunctional voiding.

### Methods

This double-blind, randomized controlled trial will employ a parallel-group design and will be conducted at the Physiotherapy Clinic of Iran University of Rehabilitation Sciences, Tehran, Iran. The study population will comprise women aged 18–50 years diagnosed with dysfunctional voiding. This trial will enroll 28 participants, equally allocated to two parallel groups (n = 14 per group). Both groups will receive standard urotherapy and pelvic floor exercises as baseline interventions. The experimental group will additionally undergo 20-minute interferential stimulation twice weekly for 10

**Data availability statement:** No datasets were generated or analysed during the current study. All relevant data from this study will be made available upon study completion.

**Funding:** This research was founded by the deputy of research, school of rehabilitation, Iran University of Medical Sciences (IUMS). The funders had no role in the study design, data collection, decision to publish, or preparation of the manuscript. Initials of the authors who received each award: SSBZ Grant numbers: 1403-1-75-28251 URL: https://research.iums.ac.ir/.

**Competing interests:** The authors have declared that no competing interests exist.

sessions under clinical supervision. Primary outcomes include maximum urine flow rate (Qmax) and severity of lower urinary tract symptoms, assessed at three times: before treatment, after treatment, and after a three-month follow-up. All statistical analyses will be conducted using SPSS software (version 26), with statistical significance set at $p < 0.05$,

## Expected results

It is hypothesized that using interferential stimulation will have a positive effect on clinical symptoms and urodynamic parameters in women with dysfunctional voiding.

## Trial registration

This trial is registered in the Iranian Registry of Clinical Trials (IRCT) under the registration number IRCT20180611040061N2.

---

## Introduction

According to the International Continence Society (ICS), voiding dysfunction is defined as abnormally slow and/or incomplete micturition, diagnosed through symptoms and urodynamic investigations (including abnormal urine flow rates and/or abnormally high post-void residuals [PVR]) [1]. Its prevalence in women ranges from 6.8% to 61.7%, increasing with age [2]. Symptoms typically manifest during the voiding phase or immediately after voiding, with the most common reported complaints being the sensation of incomplete bladder emptying and weak urine flow. This dysfunction may also present alongside storage phase symptoms, as their underlying pathophysiology can develop independently or in combination [3–5]. In a normal voiding process, an initial voluntary decrease in urethral pressure occurs due to relaxation of the urethra and surrounding striated muscles. This is followed by a sustained contraction of the detrusor muscle, facilitating bladder evacuation within a specified timeframe [4]. Consequently, the normal voiding process produces a rapid and unobstructed flow with no post-void residual volume due to urethral relaxation without increased detrusor pressure [4].

The causes of voiding dysfunction are generally impaired detrusor contractility (reduced activity or complete loss) or bladder outlet obstruction (BOO) [3]. Prospective study identify that BOO represent the most common etiology of voiding dysfunction [4]. BOO is characterized by a condition in which, despite the absence of obvious infection or injury, detrusor pressure increases and urine flow rate decreases during voiding. This condition may be accompanied by increased PVR volume [4,6]. BOO in women can arise from structural or functional causes. Major functional causes include dysfunctional voiding and detrusor-sphincter dyssynergia [3]. Dysfunctional voiding is characterized by intermittent and/or fluctuating urine flow rate caused by involuntary periurethral striated muscle contractions during voiding in neurologically normal individuals [1].The definitive diagnosis of all types of voiding dysfunction requires both clinical symptoms evaluation and objective

assessment through urodynamic studies [3]. PVR volume and bladder emptying efficiency are two clinical variables that can be used to evaluate voiding dysfunction, particularly obstructive types, and to monitor treatment response [7]. A maximum urine flow rate (Qmax) <12 mL/s and a PVR > 10% are considered reliable diagnostic indicators for BOO [8]. The principal objectives in managing voiding dysfunction are to restore normal voiding function and prevent secondary complications. Due to the lack of acceptable effectiveness of medical treatments, alternative and complementary treatments like rehabilitation are being developed. One of the conservative methods for the initial management of voiding dysfunction is urotherapy to stimulate bladder emptying and improve the sensation of bladder fullness. This method involves proper toilet training and a regular voiding schedule. In patients who are resistant to urotherapy, clean intermittent catheterization is recommended to prevent complications such as urinary tract infections. However, this approach is temporary and does not improve voiding function [9]. Currently, surface electrical stimulation is widely used to treat various pelvic floor disorders [9,10]. Studies demonstrate its positive effects on functional bladder capacity by promoting relaxation of pelvic floor muscles and modulating bladder pressure across various lower urinary tract disorders [11]. The mechanism of action for these currents in patients with voiding dysfunction involves enhancing afferent activity from the urethral sphincter to the spinal cord while improving bladder sensation and reducing detrusor muscle inhibition [12]. Additionally, intermittent contractions of the pelvic floor muscles contribute to improved coordination in bladder emptying [13]. Interferential stimulation is one type of the surface electrical stimulation that is effective for urinary and fecal disorders, such as urinary incontinence and constipation [14–16]. It consists of two out-of-phase sinusoidal currents with a medium frequency (1–10 kHz). These high frequency currents reduce skin resistance, generating low-frequency interference currents in deep tissues with effects similar to low-frequency stimulation [17]. Given that the pelvic floor muscles play a crucial role in sacral reflexes, stimulation of this area using low-frequency interferential current enhances the efferent pathways of the pelvic floor and influences bladder function at the sacral level [13].

Despite multiple clinical studies have investigated the effects of interferential stimulation on pelvic floor disorders like urinary incontinence and constipation [14–16], a comprehensive review failed to identify any studies examining the effects of interferential stimulation in women with any voiding dysfunction. Recent investigations have focused on its effects on children experiencing similar conditions [9,11]. Thus, this double-blind randomized clinical trial is designed to evaluate the effectiveness of interferential stimulation on lower urinary tract symptoms and uroflowmetry findings in women with dysfunctional voiding. It is hypothesized that interferential stimulation will improve clinical symptoms as well as uroflowmetry parameters in this population.

This protocol study aims to investigate and compare the effects of interferential electrical stimulation and standard urotherapy treatment on clinical symptoms and urodynamic findings in women with dysfunctional voiding.

## Materials and methods

### Study design

This study protocol for a double-blind randomized clinical trial (RCT) including a control group with 2 parallel groups is registered in the Iranian Registry of Clinical Trials (IRCT) under registration number IRCT20180611040061N2. Randomization in this study will be conducted using opaque sealed envelopes. The study will be performed at the Physiotherapy Clinic, Faculty of Rehabilitation, Iran University of Medical Sciences, Tehran, Iran. The study will follow the SPIRIT Checklist for reporting protocol [18]. Participants will be recruited from December 2024 to December 2025.

### Recruitment

Sampling will be conducted using a non-probability convenience sampling method, based on specialist referrals and inclusion criteria. Participants will be recruited from women residing in Tehran.

## Informed consent

All participants will be informed of the goals and procedures of the study. They will be assured that if any treatment demonstrates superiority over others at the end of the study, the superior treatment will be provided to the other group. Written informed consent will be obtained from all eligible participants prior to the commencement of the study. The informed consent form has been developed in accordance with the Ethics Committee of Iran University of Medical Sciences (IUMS) guidelines. Participants may withdraw from the study at any time without penalty. This study has received ethical approval from the Ethics Committee of IUMS (Code: IR.IUMS.REC.1403.547) and will adhere to the principles of the Helsinki Declaration. All personal information of participants will be maintained with strict confidentiality. The participant schedule is presented in Figure 1.

| Time point | Enrolment | Allocation | Post-allocation | | |
|---|---|---|---|---|---|
| | $-T_1$ | $T_0$ | $T_1$ | $T_2$ | $T_3$ |
| **Enrolment:** | | | | | |
| Eligibility screen | × | | | | |
| Informed consent | | × | | | |
| Demographic Questionnaire | | × | | | |
| Allocation | | × | | | |
| **Interventions:** | | | | | |
| Baseline plan+ Real interferential | | | × | | |
| Baseline plan+ Sham interferential | | | × | | |
| **Assessments:** | | | | | |
| Uroflowmetry | | × | | × | × |
| ICIQ-FLUTS | | × | | × | × |
| PFDI-20 | | × | | × | × |
| Bladder diary | | × | | × | × |
| GRCS | | | | × | × |

**Fig 1. Participant schedule.**

## Eligibility criteria

The inclusion criteria include: 1) women aged between 18 and 50 years, 2) positive urodynamic criteria supporting dysfunctional voiding diagnosis (Qmax ≤12 mL/s, detrusor pressure at maximum urinary flow rate ≥ 20 cm $H_2O$, and active EMG during voiding phase, 3) urologist-confirmed dysfunctional voiding diagnosis. The exclusion criteria include: 1) pregnancy and breastfeeding, 2) a history of spinal or pelvic surgery within past 2 years, 3) a history of back or pelvic malignancies within past 2 years, 4) structural, neurological, and chronic disorders, 5) neurogenic bladder, 6) voiding dysfunction secondary to medication and diseases, 7) pelvic organ prolapse > Stage II, 8) Active pelvic infection/pathology during the treatment process, 9) impossibility of necessary patient cooperation in the treatment process, 10) refusal to participate [3,8,9].

## Outcomes

The primary outcomes in this study will be the maximum urine flow rate (Qmax) and the severity of lower urinary tract symptoms. The secondary outcomes will include: urinary flow patterns, average flow rate (Qave), micturition duration, voided volume, flow time, Detrusor pressure at maximum flow (Pdet@Qmax), 24-hour voiding frequency, the pelvic floor distress inventory-20 (PFDI-20), and patient reported global rating. Generally, the outcomes will be assessed through uroflowmetry, which will be performed by a urologist, or through questionnaires administered by a physical therapist. An evaluator will conduct all assessments for both groups pre- and post-intervention. A skilled pelvic floor physiotherapist will also provide all rehabilitation treatments for both groups.

### Primary outcome measures.

- Maximum urine flow rate (Qmax):

Urodynamic studies are considered the gold standard for diagnosing voiding dysfunction. This test provides a functional assessment of the lower urinary tract [19]. Urodynamic test, identifies voiding dysfunction when the maximum urinary flow rate (Qmax) is ≤ 12 mL/s, detrusor pressure at maximum urinary flow rate flow is ≥ 20 cm $H_2O$, and PVR volume exceeds 10% of bladder capacity [8,20]. The maximum urinary flow rate (Qmax) represents the highest flow rate achieved during micturition. Clinicians commonly use Qmax to evaluate patients with lower urinary tract symptoms. Although Qmax declines with age and is volume-dependent, abnormally reduced values may indicate BOO [19,21].

- Persian version of the international consultation on Incontinence Questionnaire – Female Lower Urinary Tract Symptoms (ICIQ-FLUTS):

This questionnaire will be used to measure the severity of lower urinary tract symptoms [22,23]. The ICIQ-FLUTS is a self-report questionnaire used in clinical practice and research to assess lower urinary tract symptoms and their impact on quality of life in women. The questionnaire comprises 12 questions, each consisting of two parts. The first part scores symptom severity using a Likert scale (0–4), where a higher score indicates greater severity. The second part assesses the perceived bother caused by each symptom, scored from 0 to 10, with higher scores indicating greater bothersomeness. The questionnaire items are divided into three categories: 1) Bladder filling phase symptoms (4 questions), 2) Bladder emptying phase symptoms (3 questions), 3) Incontinence symptoms (5 questions) [23]. The ICIQ-FLUTS will be administered before and after the intervention, and after a three-month follow-up.

### Secondary outcome measures.

- Uroflowmetry parameters

Uroflowmetry is a non-invasive urodynamic test that evaluates voiding function both before treatment and during the follow-up period of therapeutic interventions. In this test, patients void into a device that measures urine flowrate. Post-test, a urinary flow curve is generated, and the quantitative parameters are reported [24,25]. Uroflowmetry is performed

on patients with a full bladder in a private setting using standardized protocols. Using this test, the following secondary outcome measures will be recorded: urinary flow patterns, average flow rate (Qave), micturition duration, voided volume, flow time, and detrusor pressure at maximum flow (Pdet@Qmax). Abnormal urinary flow patterns include intermittent, sawtooth, or prolonged curves. A time to maximum flow exceeding 5 seconds or residual urine volume >10% may indicate voiding dysfunction [8,20,26]. The standard clinical protocol comprises two steps: uroflowmetry without catheterization, followed by post-void residual (PVR) measurement via ultrasound [19]. Uroflowmetry will be performed before and after the intervention, and after a three-month follow-up.

• Persian version of 3-day bladder diary [27]:

This is a table in which patients record voiding frequency, the volume of urine voided each time, the frequency of urinary incontinence, bladder fullness sensation, pad usage (in cases of incontinence), and intermittent catheterization (in cases of retention disorders). It also documents fluid intake (types and volumes) during a specific period. For retention disorders, it is recommended to use voiding diaries for three days or more [6,28]. In this study, the International Consultation of Incontinence Questionnaire (ICIQ) bladder diary will be utilized [27]. The diary will be completed before treatment, post-treatment, and after a three-month follow-up. Participants will record aforementioned parameters over three typical days (two weekdays and one weekend day) representing their normal routine. To facilitate accurate data collection, participants will be provided with two standardized measuring cups: one for measuring urine output and another for monitoring fluid intake. Prior to initiation, each participant will receive a comprehensive verbal explanation and written instructions detailing diary procedures. The importance of consistent documentation will be emphasized, highlighting how this documentation enhances problem recognition and enables effective treatment monitoring. For each entry, time and the following details should be provided:

In the Time column, patients should record the time whenever they use the toilet, consume fluids, go to bed, or wake up. "BED" should be noted when they go to bed and "WOKE" when they wake up. In the Drinks column, they should document the amount (in milliliters) and type of beverage consumed (e.g., water, tea, coffee). In the Urine Output column, the volume of urine should be measured (in milliliters) using a measuring jug. If they urinate but are unable to measure the amount, they should place tick (✓) in this column. If they experience any urine leakage, the option "LEAK" must be marked. In the Bladder Sensation column, they should describe how their bladder felt before urinating using the following codes:

0. No sensation of needing to urinate, but urinate for social reasons (e.g., before leaving home or due to uncertainty about toilet access), 1. Normal urge without urgency, 2. Felt urgency, but it subsided before reaching the toilet, 3. Felt urgency and reached the toilet in time (still urgent but no leakage), 4. Felt urgency but could not reach the toilet in time (leakage occurred).

In the Pads Column, if a pad is used or changed, a tick (✓) should be marked in this column [27].

Persian version of pelvic floor distress inventory-20 (PFDI-20) [29]

The PFDI-20 questionnaire is highly recommended by the International Continence Society, assesses pelvic floor disorders [30]. This 20-item questionnaire evaluate the severity of pelvic floor disorders across three subscales. Each item scored from 0 to 4 based on the severity of symptom. Higher scores indicate greater symptom severity. The 3 subscales of this questionnaire are as follows: 1) Pelvic Organ Prolapse Distress Inventory (POPDI) with 6 items 2) Colorectal-Anal Distress Inventory (CRADI) with 8 items 3) Urinary Distress Inventory with 6 items [31,32]. In this study, patients will complete this questionnaire before treatment, post-treatment and after a three-month follow-up.

• Global rating of change score (GRCS):

This Patient-reported outcome measures (PROMs) quantify symptom evolution over time, typically assessing treatment efficacy or disease progression. GRCS uses a single-item 7-point scale where patients rate symptom change: 0 indicates no change, negative values (−1 to −3) indicate deterioration, and positive values (+1 to +3) indicate improvement [33]. Participants will complete the GRCS post-treatment and after a three-month follow-up.

## Treatment protocol

- Similar basic treatment in both groups: All participants will receive standardized pharmacotherapy per consulting urologist prescriptions. Each group will undergo routine urotherapy and exercise therapy for voiding dysfunction. The urotherapy program, delivered a pelvic floor specialist physiotherapist who provides education on urinary/defecatory tract function, optimal fluid intake, scheduled voiding (every 2–3 hours), proper toilet habits, and intermittent catheterization when indicated [11]. Proper spinal alignment is emphasized during toilet retraining. Patients are instructed to widen their thighs, relax abdominal and pelvic floor muscles, and maintain a continues urinary flow during voiding. All patients will be encouraged to maintain regular bowel habits, like their voiding schedule, and to consume a high-fiber diet [9]. Exercise therapy will begin by teaching pelvic floor muscles anatomy and function via visual aids and palpation. For correct pelvic floor muscles recruitment, the skilled physiotherapist will instruct patients to contract the pelvic floor muscles as if interrupting urination. During the first session, the physiotherapist will initially assess proper pelvic floor contractions through superior perineal movement observation Supplemental verification will be conducted via external palpation when visual assessment proves insufficient to confirm correct muscle engagement [34]. Patients will also be instructed to breathe normally during exercises and to avoid using their abdominal and gluteal muscles. After three weeks, and once it is confirmed that patients have learned the correct method for contracting the pelvic floor muscles, exercises will be progress to various functional positions to coordinate the pelvic floor muscles with surrounding muscle groups [10,35,36]. The exercise protocol comprises diaphragmatic breathing, pelvic floor muscle training (10-second contractions alternating with 30-second relaxation of the pelvic floor muscles), and stretching exercises [9,13,37]. All participants will perform the prescribed exercises twice daily (10 minutes per session) at home. Full urotherapy and exercise protocols are detailed in Table 1.

- Interferential stimulation in the treatment group: Participants in the treatment group will receive 20 minutes of interferential current using a Novin device (2 Novin Medical Equipment Co., Isfahan, Iran) delivering a 4000 Hz carrier frequency and a 80–160 Hz beat frequency [11,13,31]. Four 5×5 cm self-adhesive electrodes will be used to deliver the current to the skin surface. The electrodes will be arranged crosswise, with one electrode from each channel positioned on the pubic symphysis and the other electrodes placed reciprocally on the ischial tuberosities to ensure effective current flow through the pelvic floor [11,13]. Intensity will be gradually increased to strong but comfortable paresthesia without pain. Adjustments will be made immediately if discomfort occurs. Interferential current consists of 20-minute sessions applied twice weekly for 10 sessions concurrently with urotherapy, administered at the physiotherapy clinic of the Iran University Rehabilitation Faculty.

**Table 1. Therapeutic Exercise and Urotherapy Protocol.**

| Component | Details | Duration/ frequency |
|---|---|---|
| **Routine Urotherapy** | - Education on urinary/defecatory function, fluid intake, scheduled voiding (every 2–3 hours), proper toilet habits.<br>- Emphasis on spinal alignment, thigh widening, and relaxation of abdominal/pelvic muscles during voiding.<br>- High-fiber diet and regular bowel habits.<br>- Foley catheter instruction if needed. | **Home program:**<br>Every day<br>**Clinic:**<br>10 sessions (twice weekly) |
| **Exercise Therapy** | • Diaphragmatic breathing and coordination exercises<br>• **Pelvic Floor Muscle Training**:<br> - *Weeks 1–3*: 10-sec contraction + 30-sec rest,. Positions: Supine (first week), sitting (second week) and side lying (third week).<br> - *Weeks 4–5:* 10-sec contraction + 30-sec rest, Positions: Progress to standing and functional training (e.g., tilt board and ball exercise).<br>• **Stretching**: Adductor, piriformis, gluteal, hamstring muscles (progressively added). | **Home program**:<br>10 reps/set, 2x/day<br>**Clinic**:<br>10 sessions (twice weekly) |

- Interferential stimulation in the control group: The control group will receive placebo interferential electrical stimulation using the same device, settings, and electrode placement as the treatment group. Current intensity will be adjusted to the patient's initial sensory threshold and terminated after one minute [38,39].Electrodes will remain attached for the entire 20-minute session duration without active current delivery, preserving consistent experimental conditions across groups.

### Randomization

In this study, patients will be randomly assigned to treatment and control group in a 1:1 ratio using a computer-generated random number sequence with block randomization (block size: 4).

### Allocation concealment

Allocation concealment refers to the process of concealing the allocation sequence from participants in clinical trials to reduce selection bias [40]. In this study, allocation concealment will be implemented using opaque, sealed envelopes containing group assignments organized in blocks of 4. Both the preparation of the envelopes, and the selection process after patient admission will be performed by independent individuals are not involved in the study. Thus, participants and therapists participating in the study remain unaware of the group assignment until the envelope is opened.

### Blinding

In this study, both the patients and the specialist evaluating the treatment outcomes will be blinded to group assignment. To avoid the placebo effect of interferential stimulation and selection bias, a sham current will be applied and patients will remain blinded to their group assignment [38,41]. Various methods have been used for this purpose in previous studies. Recent studies have applied interferential current for a short treatment period and then stopped the current during the rest period, while maintaining identical electrode placement method across both groups [38,39]. In our study, to maintain patient blinding, the same electrode placement method will be used in both groups. Additionally, the control group will receive interferential current for only the first minute of treatment. To minimize the therapeutic effects in the control group, current intensity will be adjusted to match the initial sensations reported by patients; however, the treatment group will receive current at their maximum tolerable intensity.

To prevent assessment bias, the evaluating specialist will remain blinded [41]. Consequently, group assignment will occur after patient referral to this specialist, who will remain unaware of allocations. However, due to the required treatment adjustments in this study, it will not be possible to blind the physiotherapist responsible for the treatment.

### Sample size

In this study, the sample size calculation for maximum urine flow (Qmax) as a primary outcome was performed using G*Power 3.1.9.2 software. A total of twenty-eight subjects will be enrolled, with fourteen assigned to each of the two groups (with an expected 10% dropout rate). For the repeated measures ANOVA (three measurements, two groups), α and power (1- β) values for the sample size calculation, were set at 0.05 and 0.8, respectively. Additional parameters included the effect size, number of groups, number of measurements, and the correlation between repeated measures. The effect size was derived from a study conducted by Kajbafzadeh, which reported mean (standard deviation) Qmax values of 21 (8.3) in the intervention group and 12.8 (4.8) in the control group after a one-year follow-up [9]. Based on these values, the calculated effect size was 0.854. The analysis incorporates two groups and three measurements. Correlation values between repeated measures ranged from 0 to 0.99; ultimately, a correlation of 0.7 was selected, as it yielded the largest required sample size.

## Statistical analysis

Demographic characteristics (age and BMI) will be summarized using descriptive statistics (mean ± standard deviation). To compare these variables between the two groups, we will use independent t-test for normally distributed data and the corresponding nonparametric test otherwise. Normality will be assessed using the Shapiro-Wilk test. To evaluate outcomes over time (pre-intervention, post-intervention, and three-month follow-up), repeated-measures ANOVAs will be applied, including time (within-subject) and group (between-subject) factors, with interaction effects (Time × Group) examined. In cases of non-normality, a nonparametric equivalent will be used.

Each study group will be subjected to its own repeated measures ANOVA to investigate within-group differences over time, ensuring individual patterns of change are assessed distinctly. If a statistically significant difference in the mean values over time is found, post hoc tests with Bonferroni and Tukey corrections will be used.

At each time point, group comparisons will again use the independent t-test or its corresponding nonparametric test, depending on the data's distribution. The data will be analyzed using SPSS version 26. The significance level in this study is assumed to be 5%.

Minimum clinically important differences (MCIDs) will be addressed for each tool to evaluate clinically significance. For ICIQ-FLUTS an anchor-based CID of approximately 3.4 points will be considered. Although the reference study population consisted of women with stress incontinence, we applied this threshold with appropriate caution given the overlapping voiding domains relevant to our sample [42]. For PFDI-20, the MCID of 45 points reported by Barber et al. in a surgical cohort is available [43]. Although our population will receive non-surgical treatment, this remains the most widely accepted MCID in the literature. The GRCS was not used as an outcome measure, but rather as an anchor to estimate MCID for the ICIQ-FLUTS and PFDI-20, following the original methodology proposed by Jaeschke et al. [44]. We defined "a little better" on the GRCS as the threshold for minimal important change. If methodologically feasible, sample-specific anchor-based MCIDs may be derived using GRCS response categories.

## Discussion

The present study aims to investigate the effectiveness of interferential stimulation on clinical and uroflowmetry findings in women with voiding dysfunction. To the best of our knowledge, this represents the first double-blind, randomized controlled clinical trial evaluating interferential current efficacy in this population. Evidence from pediatric studies suggests that combining interferential stimulation with standard urotherapy significantly improves voiding parameters in children with underactive bladder, including increased voided volume, voided frequency, and reduced voiding time [45].

The bladder voiding cycle involves coordinated activity between afferent fibers, central nervous system (CNS) integration, efferent fiber signaling, and detrusor muscle contraction. Evidence suggests that electrical stimulation may modulate activity at multiple levels of both the peripheral and central nervous systems, potentially restoring the balance between excitatory and inhibitory regulatory mechanisms [9]. Moreover, electrical currents can induce reflex inhibition of the pelvic nerves, thereby increasing bladder capacity. By stimulating the afferent pathways of the pudendal nerve, these electrical currents activate the efferent pathways of the hypogastric nerves, which consequently reduce sympathetic activity and decrease involuntary the pelvic floor muscles contractions [13].

During bladder expansion, myelinated Aδ-afferent fibers in the bladder wall transmit rapid signals to initiate voiding and detrusor muscle contraction. Evidence suggests that electrical stimulation provides a noninvasive alternative capable of modulating peripheral nerves to achieve comparable bladder activation and voiding effects [10,46]. Interferential stimulation has proven effective in enhancing pelvic muscle strength and restoring neuro-urinary reflexes, indicating that it may improve bladder emptying through neuromodulatory mechanisms. Its therapeutic arises from the intersection of two medium-frequency currents within target tissues, generating an amplitude-modulated pulsating effect. Compared to conventional stimulation, medium-frequency currents provide distinct advantages: 1) reduced tissue impedance, 2) deeper penetration, and 3) improved patient tolerance due to decreased cutaneous discomfort [9]. Pediatric studies show

combined interferential stimulation and standard urotherapy significantly improve voiding parameters, increased voided volume, and frequency and reduced voiding time in children with underactive bladder [45].

If interferential stimulation demonstrates significant improvements in clinical symptoms and uroflowmetry parameters in women with voiding dysfunction, this non-invasive treatment could be recommended as an adjunct to other therapeutic approaches for symptom management.

## Adverse effects

Interferential stimulation penetrates deeply with less discomfort compared to other forms of electrical currents. Also, it is easy to apply. The most methodologically similar trials have documented no serious adverse events [9,47].

## Data monitoring

As this study involves minimal-risk interventions, an independent data monitoring committee will not be necessary.

## Supporting information

**S1 Checklist.  SPIRIT checklist study protocol.**
(DOCX)

**S1 File.  Registered protocol for ethics code.** English Translate.
(PDF)

**S2 File.  Registered protocol for ethics code.**
(PDF)

## Author contributions

**Conceptualization:** Seyedeh Saeideh Babazadeh-Zavieh, Sona Tayebi.

**Formal analysis:** Niloofar Rabiei.

**Investigation:** Fahimeh Karshenas, Seyed Mohammad Jafar Haeri.

**Methodology:** Seyedeh Saeideh Babazadeh-Zavieh, Fahimeh Karshenas, Sona Tayebi, Niloofar Rabiei.

**Project administration:** Seyedeh Saeideh Babazadeh-Zavieh.

**Supervision:** Seyedeh Saeideh Babazadeh-Zavieh, Seyed Mohammad Jafar Haeri.

**Writing – original draft:** Seyedeh Saeideh Babazadeh-Zavieh, Fahimeh Karshenas, Niloofar Rabiei.

**Writing – review & editing:** Seyedeh Saeideh Babazadeh-Zavieh, Seyed Mohammad Jafar Haeri.

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
