## [Decision Letter · Decision Letter 0]

9 Jun 2025

PONE-D-25-21061Effects of Interferential Stimulation on Clinical Symptoms and Urodynamic Findings in Women with Voiding Dysfunction: A Protocol of Randomized Clinical TrialPLOS ONE

Dear Dr. Haeri,

Thank you for submitting your manuscript to PLOS ONE. After careful consideration, we feel that it has merit but does not fully meet PLOS ONE’s publication criteria as it currently stands. Therefore, we invite you to submit a revised version of the manuscript that addresses the points raised during the review process.

We look forward to receiving your revised manuscript.

Kind regards,

Mazyar Zahir

Academic Editor

PLOS ONE

4. Please ensure that you refer to Figure 1 in your text as, if accepted, production will need this reference to link the reader to the figure.

Reviewers' comments:

Reviewer's Responses to Questions

**Comments to the Author**

1. Does the manuscript provide a valid rationale for the proposed study, with clearly identified and justified research questions?

Reviewer #1: Yes

Reviewer #2: Yes

Reviewer #3: Yes

2. Is the protocol technically sound and planned in a manner that will lead to a meaningful outcome and allow testing the stated hypotheses?

Reviewer #1: Yes

Reviewer #2: No

Reviewer #3: Yes

3. Is the methodology feasible and described in sufficient detail to allow the work to be replicable?

Reviewer #1: Yes

Reviewer #2: No

Reviewer #3: No

4. Have the authors described where all data underlying the findings will be made available when the study is complete?

Reviewer #1: Yes

Reviewer #2: Yes

Reviewer #3: Yes

5. Is the manuscript presented in an intelligible fashion and written in standard English?

Reviewer #1: Yes

Reviewer #2: Yes

Reviewer #3: Yes

6. Review Comments to the Author

You may also provide optional suggestions and comments to authors that they might find helpful in planning their study.

Reviewer #1: The authors reported a study aiming to describe the effects of inferential stimulation on clinical symptom and urodynamic findings in women with voiding dysfunction. Generally the study is interesting, but it has some issues. Below there are some comments which need to be taken into account during the revision of this manuscript.

1. Please define voiding dysfunction according to the ICS guidelines.

2. The VD is a general term. Which type of VD patients will include in the study? Describe it completely.

3. Please define positive urodynamic criteria favor the voiding dysfunction diagnosis in the inclusion criteria part.

4. Please include these criteria as exclusion criteria “ 5) absence of pregnancy and breastfeeding, 6) no history of spinal and pelvic surgery in the last 2 years, 7) no history of back and pelvic malignancies in the last 2 years, 8) absence of structural disorder, neurological disorder, history of chronic disease [1], 9) absence of neurological bladder [1], 10) absence of voiding dysfunction secondary to drug use or other diseases [1], 11) absence of high degrees of pelvic organ prolapse (degrees higher than 2)”.

5. Why do you will concern only on Qmax measurement in this study? What about other parameters of uroflowmetry such as voiding volume, Qave, voiding time ?

6. What is the purpose of using these parameters” a carrier frequency of 4000 Hz and a beat frequency of 160-80 Hz” of IF therapy in this study? Which condition will be improved? Please add a reference.

7. Will you use sham stimulation in the control group? If yes, why the output intensity will be adjusted to the patient's initial sensation threshold and the device output will be terminated after one minute? Is increasing of intensity in sham stimulation correct?Please clear it.

Reviewer #2: Statistically the manuscript is well designed and the analysis is simple. There are a few clarifications needed.

1. The sample size section is incomplete?

In the sample size section, what effect size is being used for the intervention comparison? How did one arrive at 14 per group? In the sample size calculation method in the supplementary protocol there is even less information justifying 14 per group.

2. In the Statistical analysis section, a Two way repeated measure ANOVA and one way ANOVA are mentioned. Why can’t one get the group (intervention) and time affects from the repeated measures ANOVA and test for interaction as well?

3. Unless I am misunderstanding the primary endpoints, it appears that there are about three primary outcome measures (urinary flow patterns, maximum urinary flow rate, and duration of urination). There may actually be four as per the supplementary protocol. What adjustment is being made for multiple comparisons? This will affect the overall type I error.

4. The document is nicely written in general. There are a few typos. Please check for typos:

e.g. line 415 , in the ‘Adverse’ Events section, “Also, it is easy to application”. Should be, “Also, it is easy to apply.”

Also, the sentence structure may be awkward in a number of places. Thanks!

Reviewer #3: Thank you for submitting this protocol. I don't have any major concerns about the grammar or flow of the manuscript - before resubmitting, check the capitalization of your affiliations. There are a few areas that need a bit more detail to help with my understanding:

1. Under Study Design (line 141), it states that "Patients will be recruited through urologist referrals". However, under Recruitment (line 146-47), it states that "Sampling will be conducted using a non-probability convenience sampling

method, based on specialist referrals and public announcements on social media." Please clarify how participants are being recruited, so it is consistent between the two sections.

2. Outcomes section

a. I recommend including the minimum clinically important difference (MCID) for the ICIQ-FLUTS, PFDI-20, and GRCS. These values will help to determine if the changes noted with the intervention are clinically meaningful.

b. Please provide a bit more detail about how the bladder diary will be completed. For example, how will participants measure voiding volumes and fluid intake? Which three days will be selected to complete the bladder diary (e.g., weekdays versus weekends)? What will be done to help with fully completing the bladder diaries (I often find patients have difficulties filling them out for the full three days)?

3. Treatment protocol section

a. Please provide more details about how the physiotherapist will guide pelvic floor contractions (lines 275-277), will they be using ultrasound, external palpation, or an internal exam?

b. Please clarify the pelvic floor muscle training parameters - is the participant expected to to one set of 10 in each of the three positions?

c. Will the control group remain connected to the IFC for the full 20 minutes, but with the current disconnected after 1 minute? Please clarify in lines 303-307 if the duration attached to the machine will be the same as the treatment group.

4. Statistical analysis

a. To make it clear to the reader, please list the demographic quantitative variables and the qualitative variables in the first paragraph of this section.

5. References

a. I'd recommend referring to the most recent publications as possible (e.g., Incontinence 7th Edition - https://www.ics.org/education/icspublications/icibooks) when developing your inclusion criteria and describing the urodynamic testing you'll be using.

7. PLOS authors have the option to publish the peer review history of their article (what does this mean? ). If published, this will include your full peer review and any attached files.

**Do you want your identity to be public for this peer review?** For information about this choice, including consent withdrawal, please see our Privacy Policy .

Reviewer #1: **Yes: ** Lida Sharifi- Rad

Reviewer #2: No

Reviewer #3: No

---

## [Author Response · Author response to Decision Letter 1]

7 Jul 2025

Reviewer #1: The authors reported a study aiming to describe the effects of inferential stimulation on clinical symptom and urodynamic findings in women with voiding dysfunction. Generally the study is interesting, but it has some issues. Below there are some comments which need to be taken into account during the revision of this manuscript.

We sincerely appreciate your valuable comments. All suggested revisions have been carefully incorporated and are highlighted in green for your reference.

1. Please define voiding dysfunction according to the ICS guidelines.

-Response: We appreciate your thoughtful consideration. This definition has been incorporated into both the introduction (lines 60-63) and the Abstract (line 22) of the manuscript.

2. The VD is a general term. Which type of VD patients will include in the study? Describe it completely.

-Response: We appreciate your thoughtful review. The manuscript now includes a comprehensive description of dysfunctional voiding (lines 24-27 in Abstract, and lines 85-87 in main text), which has been systematically incorporated throughout the text where relevant.

3. Please define positive urodynamic criteria favor the voiding dysfunction diagnosis in the inclusion criteria part.

- Response: We appreciate your thoughtful consideration. This criterion has been incorporated into the eligibility criteria section in Lines 167-169: (Q max equal to or less than 12 milliliters per second, a detrusor pressure at the maximum urinary flow rate equal to or more than 20, and active EMG in voiding phase).

4. Please include these criteria as exclusion criteria “ 5) absence of pregnancy and breastfeeding, 6) no history of spinal and pelvic surgery in the last 2 years, 7) no history of back and pelvic malignancies in the last 2 years, 8) absence of structural disorder, neurological disorder, history of chronic disease [1], 9) absence of neurological bladder [1], 10) absence of voiding dysfunction secondary to drug use or other diseases [1], 11) absence of high degrees of pelvic organ prolapse (degrees higher than 2)”.

- Response: We sincerely appreciate your valuable feedback. The recommended criteria have now been incorporated into the exclusion criteria section (Lines 171-178), with all necessary textual revisions implemented.

5. Why do you will concern only on Qmax measurement in this study? What about other parameters of uroflowmetry such as voiding volume, Qave, voiding time ?

-Response: Thanks for your valuable recommendation. While patients with dysfunctional voiding typically maintain normal voided volumes unaffected by treatment, we fully acknowledge the value of comprehensive urodynamic assessment. While maximum urinary flow rate (Qmax) serves as the primary outcome, we appreciate this insightful comment and confirm that our assessment protocol includes multiple secondary outcome measures (Urinary flow patterns, average flow rate (Qave), micturition duration, voided volume, flow time, Pdet@Qmax) as detailed in the Outcome Measures section (Lines182-183, and 229-231).

6. What is the purpose of using these parameters” a carrier frequency of 4000 Hz and a beat frequency of 160-80 Hz” of IF therapy in this study? Which condition will be improved? Please add a reference.

- Response: We appreciate your thoughtful consideration. This protocol represents the most frequently employed and evidence-based approach for pelvic floor disorders, as established by consensus across studies investigating interferential current therapy - including those focusing on voiding dysfunction (The following references). we added the references to this sentence. line 338.

-Sharifi-Rad L, Seyedian S-SL, Fatemi-Behbahani S-M, Lotfi B, Kajbafzadeh A-MJJoPU. Impact of transcutaneous interferential electrical stimulation for management of primary bladder neck dysfunction in children. 2020;16(1):36. e1-. e6.

-Sharifi-Rad L, Ladi-Seyedian S-S, Kajbafzadeh A-MJUJ. Interferential Electrical Stimulation Efficacy in the Management of Lower Urinary Tract Dysfunction in Children: A Review of the Literature. 2021;18(5).

7. Will you use sham stimulation in the control group? If yes, why the output intensity will be adjusted to the patient's initial sensation threshold and the device output will be terminated after one minute? Is increasing of intensity in sham stimulation correct?Please clear it.

- Response: We appreciate your comment. Yes, sham stimulation will be used in the control group. To mitigate the placebo effect of interferential electrical current and selection bias, sham current and patient blinding to the group allocation will be implemented. Various methods have been employed in previous studies to address this purpose while using electrical currents. Recent studies, in addition to using identical electrode placement in both groups, have applied interferential current for a short period during treatment and then discontinued the current. In the present study, to blind patients to the type of current used interferential electrical current will be applied for the initial 1 minute of treatment in the control group. To minimize the therapeutic effects of this modality, the current intensity will be adjusted to the initial sensation threshold of patients in the control group; however, the treatment group will receive current at an intensity tolerable for the individual. According to literatures, interferential current should be applied for at least 20 minutes with proper intensity for effectiveness. So, exposure to this stimulation for this short period of time and this inappropriate intensity will cause no bias. lines 343 and 351.

- Mendonça Araújo F, Alves Menezes M, Martins de Araújo A, Abner dos Santos Sousa T, Vasconcelos Lima L, Ádan Nunes Carvalho E, et al. Validation of a New Placebo Interferential Current Method: A New Placebo Method of Electrostimulation. Pain Medicine. 2017;18(1):86-94.

- Day SJ, Altman DG. Statistics notes: blinding in clinical trials and other studies. BMJ (Clinical research ed). 2000;321(7259):504.

- Rakel B, Cooper N, Adams HJ, Messer BR, Frey Law LA, Dannen DR, et al. A new transient sham TENS device allows for investigator blinding while delivering a true placebo treatment. The journal of pain. 2010;11(3):230-8.

- Watson T. Interferential Therapy (IFT). 2015.

Reviewer #2: Statistically the manuscript is well designed and the analysis is simple. There are a few clarifications needed.

We sincerely appreciate your valuable comments. All suggested revisions have been carefully incorporated and are highlighted in yellow for your reference

1. The sample size section is incomplete?

In the sample size section, what effect size is being used for the intervention comparison? How did one arrive at 14 per group? In the sample size calculation method in the supplementary protocol there is even less information justifying 14 per group.

- Response: We appreciate your thoughtful consideration. In this study, the sample size of maximum urine flow (Qmax) as a primary outcome was calculated with G*Power 3.1.9.2 software. Twenty-eight subjects will be included in this trial, fourteen subjects in two of the groups (with an expected 10% dropout rate). α and power (1- β) values for the sample size calculation, using repeated measures analysis of variance (ANOVA) for three measurements in two groups, were set at 0.05 and 0.8, respectively. Additional parameters factored into the sample size calculation included the effect size, number of groups, number of measurements, and the correlation between repeated measures. The effect size was derived from a study conducted by Kajbafzadeh, which reported mean (standard deviation) Qmax values of 21 (8.3) in the intervention group and 12.8 (4.8) in the control group after a one-year follow-up . Based on these values, the calculated effect size was 0.854. The number of groups and measurements used in the analysis are two and three, respectively. Correlation values between repeated measures ranged from 0 to 0.99; ultimately, a correlation of 0.7 was selected, as it yielded the largest required sample size.

The complete implementation is detailed in lines 395-404 of the manuscript.

Kajbafzadeh AM, Sharifi‐Rad L, Ladi‐Seyedian SS, Mozafarpour SJBi. Transcutaneous interferential electrical stimulation for the management of non‐neuropathic underactive bladder in children: a randomised clinical trial. 2016;117(5):793-800.

2. In the Statistical analysis section, a Two way repeated measure ANOVA and one way ANOVA are mentioned. Why can’t one get the group (intervention) and time affects from the repeated measures ANOVA and test for interaction as well?

- Response: We appreciate your thoughtful comment. To evaluate the effects of time (before the intervention, after the intervention, and after a three-month follow-up) on measurement outcomes—while considering group as a between-subject factor—repeated measures ANOVAs will be employed, assuming the data are normally distributed. In cases of non-normality, a nonparametric equivalent will be applied. Both a within-subject factor (Time) and a between-subject factor (Group) will be included in the analysis. Interaction effects (Time × Group) will be examined to determine whether group differences influenced changes in measurements over time.

To assess the effects of time on measurement outcomes within each group, separate one-way repeated measures ANOVAs will be conducted—one for the treatment group and one for the control group. If a statistically significant difference in the mean values over time is found, post hoc tests with Bonferroni and Tukey corrections will be used.

The complete implementation is detailed in lines 414-426 of the manuscript.

3. Unless I am misunderstanding the primary endpoints, it appears that there are about three primary outcome measures (urinary flow patterns, maximum urinary flow rate, and duration of urination). There may actually be four as per the supplementary protocol. What adjustment is being made for multiple comparisons? This will affect the overall type I error.

- Response: This study has two primary outcomes: maximum urine flow (Qmax) and severity of lower urinary tract symptoms. To maintain clarity, we’ve consistently specified Qmax as the primary outcome measure wherever relevant in the manuscript and have categorized other uroflowmetry parameters in the secondary outcome measures section.

lines: 191, and 222.

During the analysis process, each outcome will be treated as an independent response, and analyses such as two-way repeated measures analysis, one-way repeated measures analysis, and independent samples t-tests will be employed. In the case of multiple comparisons, as previously noted in the statistical analysis section, Tukey and Bonferroni post hoc tests (designed to maintain the overall Type I error rate at or below a desired threshold—typically 0.05) will be applied.

4. The document is nicely written in general. There are a few typos. Please check for typos:

e.g. line 415 , in the ‘Adverse’ Events section, “Also, it is easy to application”. Should be, “Also, it is easy to apply.”

Also, the sentence structure may be awkward in a number of places. Thanks!

-Response: Thank you for your comments. The text has been revised again, and typos and grammatical corrections have been made. lines 491-492.

Reviewer #3: Thank you for submitting this protocol. I don't have any major concerns about the grammar or flow of the manuscript - before resubmitting, check the capitalization of your affiliations. There are a few areas that need a bit more detail to help with my understanding:

We sincerely appreciate your valuable comments. All suggested revisions have been carefully incorporated and are highlighted in blue for your reference.

- Thank you so much. Capitalization of affiliations were corrected. line 11

1. Under Study Design (line 141), it states that "Patients will be recruited through urologist referrals". However, under Recruitment (line 146-47), it states that "Sampling will be conducted using a non-probability convenience sampling

method, based on specialist referrals and public announcements on social media." Please clarify how participants are being recruited, so it is consistent between the two sections.

- Response: Thank you for your comment. Yes, that is correct. These two sections have been clarified and patient referrals will only be made through a specialist and according to the inclusion criteria. The revised methodology detailing this standardized referral protocol appears in the Recruitment subsection (Line 151).

2. Outcomes section

a. I recommend including the minimum clinically important difference (MCID) for the ICIQ-FLUTS, PFDI-20, and GRCS. These values will help to determine if the changes noted with the intervention are clinically meaningful.

-Response: We thank the reviewer for this helpful comment. In response, we have incorporated the minimum clinically important differences (MCIDs) for all assessment tools, with detailed justification provided in the statistical analysis section (Lines 433-448).

ICIQ-FLUTS: We cited Islam et al. (2023), who reported an anchor-based CID of approximately 3.4 points and MID values ranging from 1.3 to 3.2, depending on treatment type and duration. Although their population consisted of women with stress incontinence, we applied their thresholds with appropriate caution, given the overlapping voiding domains relevant to our sample. PFDI-20: We acknowledged the MCID of 45 points reported by Barber et al. (2005) in a surgical cohort. Although our population received non-surgical treatment, this remains the most widely accepted MCID in the literature. We highlighted this discrepancy in the revised manuscript to promote interpretive transparency. GRCS: The GRCS was not used as an outcome measure, but rather as an anchor to estimate MCID for the ICIQ-FLUTS and PFDI-20, following the original methodology proposed by Jaeschke et al. (1989). We defined “a little better” on the GRCS as the threshold for minimal important change. Additionally, where appropriate, we will estimate our own anchor-based MCID values within our sample using GRCS response categories and will include these findings in the Results section for completeness.

-Nipa SI, Cooper D, Mostafa A, Hagen S, Abdel-Fattah M. Novel clinically meaningful scores for the ICIQ-UI-SF and ICIQ-FLUTS questionnaires in women with stress incontinence. International urogynecology journal. 2023;34(12):3033-40.

-Barber MD, Walters MD, Bump RC. Short forms of two condition-specific quality-of-life questionnaires for women with pelvic floor disorders (PFDI-20 and PFIQ-7). American journal of obstetrics and gynecology. 2005;193(1):103-13.

-Kamper SJ, Maher CG, Mackay G. Global rating of change scales: a review of strengths and weaknesses and considerations for design. The Journal of manual & manipulative therapy. 2009;17(3):163-70.

b. Please provide a bit more detail about how the bladder diary will be completed. For example, how will participants measure voiding volumes and fluid intake? Which three days will be selected to complete the bladder diary (e.g., weekdays versus weekends)? What will be done to help with fully completing the bladder diaries (I often find patients have difficulties filling them out for the full three days)?

-Response: We appreciate your thoughtful consideration. A full explanation of how to fill out the diary has been added to the text. Lines 253-278.

3. Treatment protocol section

a. Please provide more details about how the physiotherapist will guide pelvic floor contractions (lines 275-277), will they be using ultrasound, external palpation, or an internal exam?

-Response: The physiotherapist will initially assess proper pelvic floor contractions through visual observation of superior perineal movement during the first instructional session. Subsequent verification will be conducted via external palpation when visual assessment proves insufficient to confirm correct muscle engagement. These sentences have been added to lines 317-321.

b. Please clarify the pelvic floor muscle training parameters - is the participant expected to to one set of 10 in each of the three positions?

-Response: We appreciate your thoughtful comment. The

---

## [Decision Letter · Decision Letter 1]

21 Jul 2025

PONE-D-25-21061R1Effects of Interferential Stimulation on Clinical Symptoms and Urodynamic Findings in Women with Voiding Dysfunction: A Protocol of Randomized Clinical TrialPLOS ONE

Dear Dr. Haeri,

Thank you for submitting your manuscript to PLOS ONE. After careful consideration, we feel that it has merit but does not fully meet PLOS ONE’s publication criteria as it currently stands. Therefore, we invite you to submit a revised version of the manuscript that addresses the points raised during the review process.

We look forward to receiving your revised manuscript.

Kind regards,

Mazyar Zahir

Academic Editor

PLOS ONE

Journal Requirements:

**Additional Editor Comments:**

Please kindly address the following minimal comments prior to the final acceptance of your manuscript:

1) Ensure the accurate use of the terms "dysfunctional voiding" and "voiding dysfunction". As noted by the first reviewer, the term "dysfunctional voiding" is primarily a pediatric diagnosis. In older patients, alternative terms such as "voiding dysfunction", "non-neurogenic detrusor sphincter dyssynergia", or "functional bladder outlet obstruction" may be more appropriate. Please ensure uniformity in definitions and terminology throughout.

2) We also recommend a final round of proofreading for grammatical and typographical accuracy. A thorough English language edit by a native speaker, as well as a formatting and layout edit would be appreciated. Additionally, kindly eliminate redundancies where possible. For example:

- In the statistical analysis section, several tests are described more than once and could be consolidated, 

- In the inclusion and exclusion criteria section, the reference numbers are intermingled with the actual criteria causing confusion. You may consider placing the references at the beginning of the section or alongside the first mention of the criteria.    

- Some subheadings such as "Objectives" (line 135), and "Participants Schedule" (line 165) may be removed to streamline the study.  

Reviewers' comments:

Reviewer's Responses to Questions

**Comments to the Author**

1. Does the manuscript provide a valid rationale for the proposed study, with clearly identified and justified research questions?

Reviewer #1: Yes

Reviewer #2: Yes

Reviewer #3: Yes

2. Is the protocol technically sound and planned in a manner that will lead to a meaningful outcome and allow testing the stated hypotheses?

Reviewer #1: Yes

Reviewer #2: Yes

Reviewer #3: Yes

3. Is the methodology feasible and described in sufficient detail to allow the work to be replicable?

Reviewer #1: Yes

Reviewer #2: Yes

Reviewer #3: Yes

4. Have the authors described where all data underlying the findings will be made available when the study is complete?

Reviewer #1: Yes

Reviewer #2: Yes

Reviewer #3: Yes

5. Is the manuscript presented in an intelligible fashion and written in standard English?

Reviewer #1: Yes

Reviewer #2: Yes

Reviewer #3: Yes

6. Review Comments to the Author

You may also provide optional suggestions and comments to authors that they might find helpful in planning their study.

Reviewer #1: I would like to thank the authors who made the necessary corrections to develop the manuscript. Only I have one question.

s it permissible to use the term of dysfunctional voiding in older women? Because pelvic floor muscle weakness or bladder neck disorders are more common in these older women, and the diagnosis of dysfunctional voiding was made by positive EMG activity during voiding..

Reviewer #2: None

XXXXXXXXXXXXXXXXXXXXXXXXXXXXXXXXXXXXXXXXXXXXXXXXXXXXXXXXXXXXXXXXXXXXXXXXXXXXXXXXXXXXXXXXXXXXXXXXXXXXXXXXXXXXXXXXXXXXXXXXXXXXXXXXXXXXXXXX

Reviewer #3: Thank you for making changes to the manuscript, addressing our concerns. I noticed some small typos, missing punctuation, and grammatical errors during my review - I'd encourage you to ask a friend or colleague not associated with this manuscript to proof read the final submission.

7. PLOS authors have the option to publish the peer review history of their article (what does this mean? ). If published, this will include your full peer review and any attached files.

**Do you want your identity to be public for this peer review?** For information about this choice, including consent withdrawal, please see our Privacy Policy .

Reviewer #1: No

Reviewer #2: No

Reviewer #3: No

---

## [Author Response · Author response to Decision Letter 2]

29 Jul 2025

I would like to express my sincere gratitude to the journal’s editor and all reviewers for their careful review.

All requested corrections have been addressed, and responses to specific questions are provided below.

Response to editor:

1.Ensure the accurate use of the terms "dysfunctional voiding" and "voiding dysfunction". As noted by the first reviewer, the term "dysfunctional voiding" is primarily a pediatric diagnosis. In older patients, alternative terms such as "voiding dysfunction", "non-neurogenic detrusor sphincter dyssynergia", or "functional bladder outlet obstruction" may be more appropriate. Please ensure uniformity in definitions and terminology throughout.

- Response: Regarding voiding dysfunction (as noted in the manuscript), this term broadly encompasses impaired detrusor contractility or bladder outlet obstruction (BOO). In women, BOO may arise from structural or functional causes, with major functional causes including dysfunctional voiding and detrusor-sphincter dyssynergia. Dysfunctional voiding is characterized by involuntary periurethral striated muscle contractions during voiding in neurologically normal individuals, resulting in an intermittent and/or fluctuating urine flow rate.

While the ICS definition of dysfunctional voiding does not specify age, we agree with the first referee’s observation that it occurs more frequently in younger individuals. Consequently, we narrowed the inclusion criteria’s age range to premenopausal women.

2. We also recommend a final round of proofreading for grammatical and typographical accuracy. A thorough English language edit by a native speaker, as well as a formatting and layout edit would be appreciated. Additionally, kindly eliminate redundancies where possible.

- Response: The manuscript was revised for typo, punctuation, grammer, and spelling. errors.

3.In the statistical analysis section, several tests are described more than once and could be consolidated,

- Response: We would like to clarify that repeated measures ANOVA was intentionally used for two different analytical goals in our study and it is the reason of the repetition in this section:

First, to evaluate the interaction between time and group, we applied a repeated measures ANOVA including both study groups in the same model. This enabled us to assess whether the treatment group differed from the control group in how their outcomes changed over time.

Second, to assess within-group changes over time, repeated measures ANOVAs were conducted separately for each study group. This allows us to explore how the outcomes evolved independently in the treatment and control groups without the influence of inter-group comparisons.

We appreciate your feedback and have revised the Statistical Analysis section to consolidate overlapping descriptions and improve clarity.

4. In the inclusion and exclusion criteria section, the reference numbers are intermingled with the actual criteria causing confusion. You may consider placing the references at the beginning of the section or alongside the first mention of the criteria.

- Response: The references in the inclusion criteria section were revised and -relocated to the end of section

5. Some subheadings such as "Objectives" (line 135), and "Participants Schedule" (line 165) may be removed to streamline the study.

- Response: these subheadings were removed.

Response to reviewers

Reviewer #1: I would like to thank the authors who made the necessary corrections to develop the manuscript. Only I have one question.

is it permissible to use the term of dysfunctional voiding in older women? Because pelvic floor muscle weakness or bladder neck disorders are more common in these older women, and the diagnosis of dysfunctional voiding was made by positive EMG activity during voiding..

- Response: Regarding voiding dysfunction (as noted in the manuscript), this term broadly encompasses impaired detrusor contractility or bladder outlet obstruction (BOO). In women, BOO may arise from structural or functional causes, with major functional causes including dysfunctional voiding and detrusor-sphincter dyssynergia. Dysfunctional voiding is characterized by involuntary periurethral striated muscle contractions during voiding in neurologically normal individuals, resulting in an intermittent and/or fluctuating urine flow rate.

While the ICS definition of dysfunctional voiding does not specify age, we agree with your observation that it occurs more frequently in younger individuals. Consequently, we narrowed the inclusion criteria’s age range to premenopausal women.

Reviewer #2: None

XXXXXXXXXXXXXXXXXXXXXXXXXXXXXXXXXXXXXXXXXXXXXXXXXXXXXXXXXXXXXXXXXXXXXXXXXXXXXXXXXXXXXXXXXXXXXXXXXXXXXXXXXXXXXXXXXXXXXXXXXXXXXXXXXXXXXXXX

Reviewer #3: Thank you for making changes to the manuscript, addressing our concerns. I noticed some small typos, missing punctuation, and grammatical errors during my review - I'd encourage you to ask a friend or colleague not associated with this manuscript to proof read the final submission.

- Response: The manuscript was revised for typographical errors, punctuation, and spelling was corrected

---

## [Editor Report · Decision Letter 2]

5 Aug 2025

Effects of Interferential Stimulation on Clinical Symptoms and Urodynamic Findings in Women with Voiding Dysfunction: A Protocol of Randomized Clinical Trial

PONE-D-25-21061R2

Dear Dr. Haeri,

We’re pleased to inform you that your manuscript has been judged scientifically suitable for publication and will be formally accepted for publication once it meets all outstanding technical requirements.

Kind regards,

Mazyar Zahir, MD

Academic Editor

PLOS ONE
---

## [Editor Report · Acceptance letter]

PONE-D-25-21061R2

PLOS ONE

Dear Dr. Haeri,

I'm pleased to inform you that your manuscript has been deemed suitable for publication in PLOS ONE. Congratulations! Your manuscript is now being handed over to our production team.

Kind regards,

on behalf of

Dr. Mazyar Zahir

Academic Editor

PLOS ONE